# Ivermectin (IVM) Possible Side Activities and Implications in Antimicrobial Resistance and Animal Welfare: The Authors’ Perspective

**DOI:** 10.3390/vetsci9010024

**Published:** 2022-01-11

**Authors:** Cristian Piras, Enrico Gugliandolo, Fabio Castagna, Ernesto Palma, Domenico Britti

**Affiliations:** 1Department of Health Sciences, Campus Universitario “S. Venuta”, University “Magna Græcia” of Catanzaro, Viale Europa, 88100 Catanzaro, Italy; palma@unicz.it (E.P.); britti@unicz.it (D.B.); 2Department of Veterinary Sciences, University of Messina, 98168 Messina, Italy; egugliandolo@unime.it; 3Nutramed S.c.a.r.l. Complesso Ninì Barbieri, Roccelletta di Borgia, 88021 Catanzaro, Italy; 4Department of Health Sciences, Institute of Research for Food Safety & Health (IRC-FISH), University of Catanzaro Magna Græcia, 88100 Catanzaro, Italy

**Keywords:** ivermectin, antimicrobial resistance, proteomics, green veterinary pharmacology (GVP), immunomodulation

## Abstract

Ivermectin has a wide number of many diverse functions. Certainly, it is irreplaceable for the treatment of parasitic pathologies in both human and veterinary medicine, and the latter represents the major field of its application. It has been called the “drug for the world’s poor” because of its role as a saviour for those living on the margins of society, in underdeveloped areas afflicted by devastating and debilitating diseases, such as Onchocerciasis and Lymphatic filariasis. It showed huge, unexpected potential as an antibacterial (*Chlamydia trachomatis* and mycobacteria), and it has antiviral and anti-inflammatory properties. The research line described here is placed right in the middle of the investigation on the impact of this drug as an antimicrobial and an immunomodulator. Being a drug widely employed for mass administration, it is mandatory to broaden the knowledge of its possible interaction with bacterial growth and its generation of antimicrobial resistance. Equally, it is important to understand the impact of these drugs on the immune systems of animal species, e.g., horses and dogs, in which this drug is often used. More importantly, could immunomodulation and antibacterial activity promote both bacterial growth and the occurrence of resistance mechanisms?

## 1. Introduction

Nature offers support to mankind with naturally occurring compounds synthesized by *Streptomyces avermitilis* and capable of significantly reducing the incidence of devastating diseases, such as river blindness and lymphatic filariasis [1,2]. *Streptomyces avermitilis* is capable of producing eight different avermectin components with structural differences [3], and among those, the B1a component has the most effective antiparasitic activity [3]. The result of this discovery was the subsequent commercialisation of the safer and more effective formulation ivermectin, which was commercialised for the veterinary, agricultural, and aquaculture market in 1981. This powerful semisynthetic mixture is composed of 80% 22,23-dihydroavermectin-B1a and 20% 22,23-dihydroavermectin-B1b, and because of its broad-spectrum anti-parasitic properties, it has been widely and massively used to counteract parasitic worms in veterinary medicine [2]. Its use in livestock quickly spread worldwide as a veterinary parasiticide for bovine, ovine, and swine application as injectable, oral, and topical formulations delivering 200–500 µ/kg of the compound. Such treatment is effective against nematodes of the gastrointestinal and respiratory tracts, conjunctival sac, and soft tissues. Its efficacy was analogously documented for arthropods, e.g., *Haematobia irritans*, the screw-worm *Chrysomyia bezziana*, lice, mange mites, and several species of ticks. In the dog, it is used for heartworm prevention at a concentration of 6 µ/kg per month during the mosquito season [4].

However, this nature-born compound is much more than just an antiparasitic drug because it shows both antibacterial and immunomodulating properties. Because of its irreplaceable use as an antiparasitic drug and widespread use worldwide, this contribution explores the other possible side effects of shaping microbial communities, possibly promoting antimicrobial resistance, and interacting with the immune system.

This research thread aims to fill the gap in knowledge of the possible role of ivermectin in inducing antibiotic resistance in bacteria and immunosuppression in host cells.

## 2. Ivermectin Effect and Antimicrobial Resistance

Avermectins belong to the macrolide family and were believed to be inactive against all bacteria. However, ivermectin, selamectin, and moxidectin have been demonstrated to be effective against mycobacterial species, including drug-resistant strains [5]. It showed a bacterial killing effect against various species of *Mycobacterium*, including *Mycobacterium tuberculosis*. Other pieces of experimental evidence demonstrated its effectiveness against *Chlamydia trachomatis* [6]. Of these two bacterial species, one responds to acid-fast staining methods, and the other is negative for gram staining. Such a difference in the staining methods of these two sensible species highlights the existing wide phylogenetic difference and poses the question about ivermectin’s possible effectiveness against a greater number of bacterial species.

## 3. Ivermectin’s Effect on Immune System

One of the main pharmacodynamic activities of ivermectin is characterised by the immunomodulation in the host [7,8]. It inhibits the proliferation of cancer cells and regulates glucose and cholesterol in animals [9]. Macrolide antibiotics, in addition to being molecules with antimicrobial activity, are useful for the management of some inflammatory diseases [10,11,12]. Chemically, ivermectin is a semisynthetic macrocyclic lactone, and like other macrolides, it blocks the lipopolysaccharide (LPS)-induced secretion of NO and prostaglandin E2 [13,14]. A quantity of 2 mg/kg produces the most efficient inhibition of the LPS-induced mortality in mice. This was observed together with an inhibition of TNF-a, IL-1b, and IL-6 production. In vitro experiments performed on RAW 264.7 cells challenged with LPS demonstrated the IVM inhibition of TNF-a, IL-1ß, and IL-6 levels [15].

## 4. Discussion

### 4.1. Antimicrobial Resistance Experimental Design

The avermectin structure is closely related to complex 16-membered macrocyclic lactones. Although they share structural features with the antibacterial macrolides and the antifungal macrocyclic polyenes, avermectins are not usually grouped with those compounds [16,17]. Chemically, the fermentation of the actinomycete *Streptomyces avermitilis* produces four homologous pairs of closely related compounds: avermectins A1, A2, B1, and B2. The four pairs are further divided into the major components A1A, A2A, B1A, and B2A. These subtle differences in the chemical structure were found to have significant functional consequences: while initial trials found that all four avermectins showed some efficacy against gastrointestinal nematodes of sheep, avermectins of the ‘B’ series showed the highest activity [18]. Further, when given orally, avermectin B1 was more active than B2, while with parenteral administration, avermectins B2 was more active than B1 [19]. Surprisingly, despite being macrocyclic lactones, their antibacterial activity is quite limited. More precisely, avermectins are reported to be very effective in killing *Mycobacterium tuberculosis*, including multidrug-resistant clinical strains, but there is no strong evidence of their activity against other bacterial species. The advantage of this class of molecules is that it is already approved for clinical and veterinary use and already has well-documented pharmacokinetic and safety profiles [5]. Lim and colleagues, on top of performing extensive literature research on the possible antibacterial activity of avermectins, performed a panel of experiments in which both Gram-positive and Gram-negative bacteria were exposed to different avermectins (doramectin, ivermectin, moxidectin, and selamectin) at concentrations as high as 256 μg/mL. The results concluded that no inhibitory effect was observed [5]. On the other hand, in the same research thread, the authors tested the inhibitory activity of avermectins against several different mycobacterial strains. The obtained results clearly demonstrated their role in inhibiting the growth of *Mycobacterium bovis* BCG and *M. tuberculosis* laboratory strains (H37Rv, CDC 1551, and Erdman) at concentrations ranging from 1 to 8 μg/mL [5].

Regarding the same topic of bacterial growth inhibition, its efficacy against *Chlamydia trachomatis* infection of epithelial cells has also been demonstrated [6].

As aforementioned, the antibacterial properties of IVM have been clearly demonstrated for mycobacterial species, but there is still a large gap in knowledge to be filled on other bacterial species. Future work on IVM should aim to clarify these scientific findings towards the additional research on Gram-positive, Gram-negative, and acid-fast bacteria behaviour after exposure to IVM at therapeutic concentrations. Such a research thread comes from the urgent need to evaluate all the possible benefits and side effects of this widely used drug. Among the side effects, the possible induction of macrolides antibiotic resistance should be investigated. This task could be accomplished by exposing one Gram-positive bacterial species (*S. aureus*), one Gram-negative bacterial species (*E. coli*), and one mycobacterium (*Mycobacterium avium* sub. *paratuberculosis*) to concentrations of IVM equal to the ones used for anthelmintic control. At the same time, the minimum inhibitory concentration of a macrolide antibiotic (MIC) should be evaluated for each species before and after IVM exposure. One possible experimental scheme to provide answers to this point is shown in Figure 1. As visible in the figure, if IVM is demonstrated to be capable of inducing resistance to macrolides, the previously saved bacterial pellets (before and after IVM exposure) will be analysed through NGS sequencing and in-depth proteomics approaches to uncover the eventual mechanisms generating antibiotic resistance [20,21]. This proteo-genomics approach will allow the analytical steps to be performed with extreme additional depth. The FASTA database necessary for the proteomics analysis will be generated specifically for each of the analysed strains.

This investigation may contribute to clarifying the potential of IVM to induce antibiotic resistance in Gram-positive, Gram-negative, and acid-fast bacteria and could uncover the molecular mechanisms adopted by mycobacterial species to survive sublethal IVM concentrations. This last point is particularly relevant considering the emerging concern regarding multidrug-resistant tuberculosis and the fact that IVM might represent one of the last possibilities to fight multidrug-resistant strains [22,23]. Label-free proteomics is very effective in the discovery of antibiotic resistance mechanisms and intrinsic resistance [24]. This has been previously demonstrated in *E.coli* isolates growing with a concentration of 10 µg/mL of enrofloxacin [21]. This evidence demonstrates that the evaluation of differential protein expression may be useful in studying resistance mechanisms. As designed, the contribution of a parallel metagenomic approach will provide a tool to improve the coverage of the proteomics dataset by reducing the gap between the analysed strains and the publicly available genomes and will also provide the possibility to search the genome for ab-resistant gene sequences.

### 4.2. Evaluation of the Effect on Immunocompetent Cells

The effect of ivermectin on immune cells has been partially studied, as aforementioned in the previous section of this manuscript. However, several questions remain unanswered, such as whether IVM is interfering/shaping the pattern of production of cytokines (i) and which pathways might be downregulated or upregulated in immunocompetent cells (ii). These mechanisms have not been elucidated yet, and this research line could contribute to trying to answer this question. Ivermectin is extensively used for parasite control both in animal production and companion animals. Despite its frequent use, there are only a few published studies documenting its effect on the animal immune system and describing its mechanism of action [9,25,26,27,28]. According to this premise, it could be productive to investigate the effect of IVM on equine and canine immune-competent cells, such as leukocytes and peripheral blood mononuclear cells (PBMC). Concerning this, a possible experimental pipeline is shown in Figure 2. Canine and equine immunocompetent cells could be challenged with LPS and exposed to ivermectin. LPS challenge would serve to evaluate the IVM immunomodulation both with and without external stimulation. Before and after LPS and IVM exposure, the pattern of secreted cytokines (i.e., IL1β, IL4, IL8, IL6, and IL10) should be analysed via commercially available kits. Subsequently, the cells can be collected and analysed via targeted gene expression by rt-PCR and bottom-up shotgun proteomics to uncover the pathways leading to the eventual immunomodulation. Previous experiments have demonstrated that it is possible to have a good proteomics coverage using 1 × 10^6^ immune cells [29] through electrospray liquid chromatography–tandem mass spectrometry after filter-aided sample preparation (FASP) [30]. The obtained results will possibly draw a complete picture of IVM effects on immune cells on the basis of both proteomics and genomics profiles.

### 4.3. Green Veterinary Pharmacology

The possible stimulation of antibiotic resistance and immunosuppressing activity in host organisms could be a negative combination if considering the long-term and massive exposure. This concern is related to the availability of a considerable biological incubator (represented in this case by the host, e.g., cows, ewes, horses, or dogs) for a large number of bacterial species that is simultaneously exposed to a macrolide-like and immunosuppressant drug. Microbiome studies [31,32,33] have clearly demonstrated that we have been underestimating the huge variety of these ecological niches. Bacterial consortia growth is analogously shaped by the status and responsiveness of the host immune system since mutual survival is the result of perfectly balanced equilibrium.

In this research line, we are trying to address (i) whether treatment with a macrolide-like molecule may promote the development of intrinsic antibiotic resistance [34] mechanisms, such as the over-expression of efflux pumps or other protection mechanisms and (ii) if there is the concrete possibility that IVM-dependent immune suppression might independently contribute to bacterial growth in this incubator.

Thus, the environmental impact of ivermectin and other drugs is also an important aspect to consider for a more conscious use of “mass administration” drugs. In this regard, we believe that more effort must be applied to improve the knowledge of “environmental pharmacology”.

These two research branches are part of a wider initiative called Green Veterinary Pharmacology, which aims to reduce the use of IVM (and other eventually used drugs) as much as possible by promoting other strategies, such as the integration of feed with non-processed crops, exploiting a similar pharmacological action. In this area, it has been recently demonstrated that a plethora of natural products can be investigated for their anthelminthic efficacy. Among these, aqueous pomegranate (*Punica granatum* L.) extract is very effective against gastrointestinal nematodes in sheep, as documented by in vitro and in vivo studies [35,36]. Such an initiative comes from the need to avoid forthcoming parasite and bacterial resistance to precious molecules, such as IVM, and provide alternative strategies for parasite control.

## Figures and Tables

**Figure 1 vetsci-09-00024-f001:**
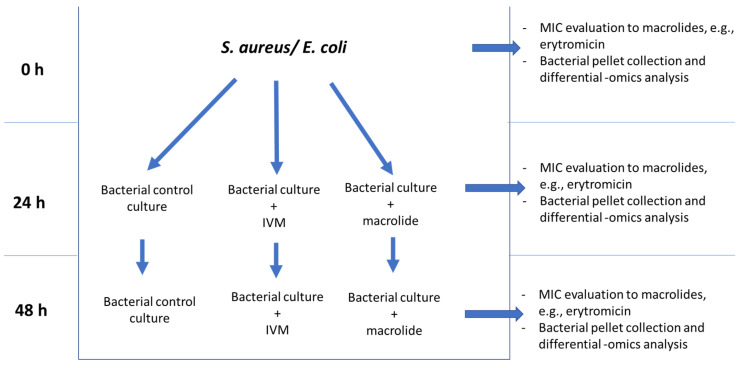
Experimental design for the evaluation of the effects of IVM therapeutic dose exposure in Gram-positive, Gram-negative, and mycobacteria.

**Figure 2 vetsci-09-00024-f002:**
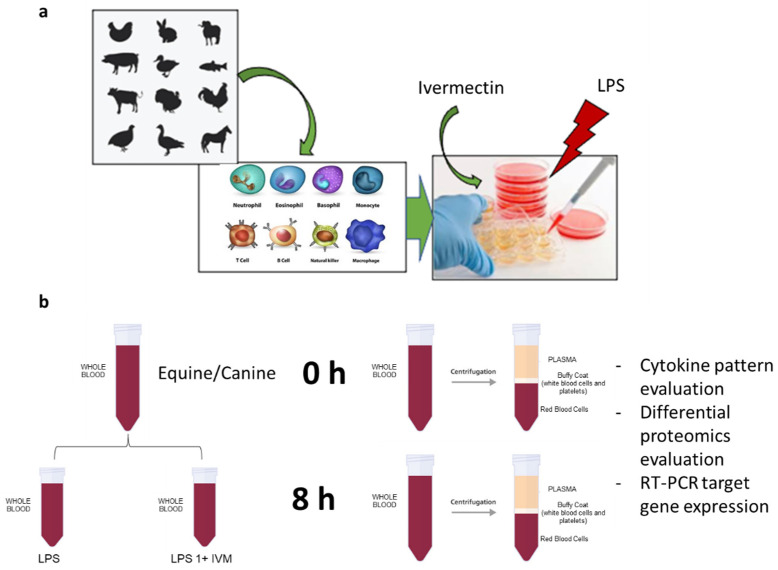
(**a**) Experimental representation for the possible evaluation of the effects of IVM on LPS-induced pro-inflammatory pathways and cytokine production. (**b**) Schematic representation of the possible workflow concerning animal blood processing to isolate immunocompetent cells.

## Data Availability

Not applicable.

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
