# Peer review of "Ivermectin (IVM) Possible Side Activities and Implications in Antimicrobial Resistance and Animal Welfare: The Authors’ Perspective"

_vetsci, 2022, doi:10.3390/vetsci9010024_

Round 1
Reviewer 1 Report
The authors present, after a brief report on the different uses of IVM, a discussion on the effect of this antiparasitic as an immunomodulator, as well as its antibacterial effects. In this last topic, they also raise the possibility that IVM can promote resistance to antimicrobials. The text is well written and clearly presents the concern of the authors in relation to the massive use of this antiparasitic drug, proposing a greater effort to improve knowledge that they called environmental pharmacology.
Author Response
We would like to thank the reviewers for their time and effort to improve the manuscript.
Considering the reviewers’ comments, we have revised the manuscript as detailed in the following response (reviewers’ comments are printed in bold) and we added additional
information accordingly.
Response to Reviewer 1: Many thanks to the referee for this comment.
Reviewer 2 Report
The manuscript describes the off-label uses of ivermectin as antibacterial, antiviral and immunomodulator agent. The manuscript needs extensive proofreadings and pay attention to grammatical errors. Furthermore, there are plenty of non-academic words and inappropriate word choice throughout the manuscript such as a lot of work (line 120), as well (lines 50 & 87) and belonging from (lines 37 & 63). The manuscript is not suitable for publication unless the abovementioned concerns and below comments are addressed.
-The study design for the induction of bacterial resistance by IVM is unclear and needs more delineation in the text to be consistent with figure 1 particularly the adding of macrolide erythromycin. Furthermore, there should be 4 groups including control (untreated bacteria), bacteria plus IVM, bacteria plus macrolide and bacteria plus IVM and macrolides. Moreover, I cannot find any correlation between the proposed experiment and its potential outcomes with anti-mycobacterial effect of IVM.
-The knowledge gap should be clearly stated in the introduction.
-Based on the context of the manuscript I would recommend to re-write the title to include the proposed proteo-genomics techniques. Remove [magic].
-The subsections should be removed from the introduction and compiled as paragraphs.
-In figure 1, what is the difference between bacterial culture and MSSA. Ivermectin also needs to be either written as IVM or complete name.
-Can authors indicate how much would be the sub-lethal doses of IVM.
-Line 16, it should be rewritten as it does not really make sense.
-Line 30, strike off [And].
-Lines 36-38, delete [once again] and re-write the whole sentences to be unambiguous.
-Gram + and – should be replaced with Gram-negative and Gram-positive throughout the manuscript.
-Line 66, replace [once] with [strains]. Remove [effective] and write down [bacterial killing effect] instead.
-Replace [that is bringing the beneficial effect] with [behind its antiviral effect].
-Lines 102-03, remove [and atypical].
-Lines 129-130, I would assume MIC here refers to macrolide therefore, a clarification is needed.
-Line 146, reference is required after [intrinsic resistance.]. intrinsic resistance.
-Line 173, replace [many] with [several].
-Line 175, change [de-regulated] with [down-regulated].
-Line 179, strike off [a].
-Line 183, [Showed] should be [shown].
- Line 184, define LPS and can authors clarify why they plan to use LPS.
-Can authors rationalise the selected time points for the all proposed experiments.
Reviewer 3 Report
The authors report what is supposed to be a review of the potential antibacterial and antiviral action of ivermectin. It is not clear what the actual aim of the study is, as the objective of the study is not defined anywhere in it.
The manuscript also includes in some of its sections a description of the design of a study they intend to carry out. Nevertheless, no results are presented and, therefore, it is no more than a mere declaration of intent.
It is not clear what the title means: all the studies published on this active ingredient and others include, in one way or another, the authors' perspective. Nor there is any magic in the pharmacological actions of ivermectin, as they have been studied and described in numerous papers for 40 years (its discoverers and developers received the Nobel Prize by their scientific -not magic- studies).
The abstract does not follow the requirements set by the journal, as it should be structured. In the abstract they indicate that it is important to understand the impact that this drug may have on the immune system of animal species such as dogs or horses, but they do not give any results (simply the design of a study to be carried out).
Line 58: I don't understand what they mean by "to explore the other possible side effects in shaping microbial communities" What does it mean “side effects" here?
Line 74-76: reference no. 7 is a review, which indicates that ivermectin inhibits cancer cell proliferation and regulates glucose and cholesterol in animals. Reference 7 does not include from which study this statement has been taken, and how they have been able to relate it to the immunomodulatory activity of ivermectin.
Lines 84-88: when talking about the anti-COVID activity of ivermectin, one must be very careful, as the WHO does not recommend its use except in the context of clinical studies that can be carried out, as those that have been carried out have many methodological limitations and a low number of patients (updated to July 2021). Reference 14 should also be updated in the bibliography (published in the journal Chest, 2021).
Any claims made about the possible antibacterial activity of ivermectin should relate to concentrations within the therapeutic range, and not only indicate concentrations obtained from in vitro studies. Micrograms should be written correctly, as µg.
Lines 144-156: It is not understood that the action of enrofloxacin is explained in such detail in a paper focused on ivermectin.
Lines 172 and 179-180: There is a certain inconsistency in what is said in line 172 (the effect of ivermectin on immune system cells is well documented), and a few lines below (179-180) it is indicated however that there are few studies documenting this effect.
Lines 223-225: the authors should indicate that the study is in vitro. Although the initiative is promising, it should then be tested in clinical practice.
Therefore, the authors should submit the manuscript for review once they have the results of the study they are going to perform.
Round 2
Reviewer 3 Report
The paper has certainly been improved by the authors. However, there are still a few minor points to be made.
- The word biocide should not be used, as it may be misleading. In the European Union, a clear distinction is made in the legislation between substances that are considered veterinary medicinal products, biocides or feed additives.
- The symbol µg has not been corrected throughout the manuscript and still appears in several places (e.g. lines 88, 170). And numbers should be separated from units (line 200, 2 mg/kg).
- I have reviewed those papers published in the journal Veterinary Sciences under the category Perspective, and they present results or review the state of the art of a certain topic. Perhaps the main problem is that the manuscript describes the proposal of a future research project to be carried out. In my experience, what I have seen throughout different journals is that authors submit a research project to certain journals for validation, and then they publish the results in that journal, but they do not publish the research project itself. I think, however, that the review is exhaustive and interesting, and opens a window for research on the role played by substances such as ivermectin in antimicrobial resistance. For this reason, they should approach the paper to be written in a more impersonal way. For example, instead of saying in line 366 "From our perspective, we will pursue the study of....", one could say "From our perspective, the study of the possible induction of macrolides antibiotics resistance in various bacterial species should be studied/assessed". And the same in line 543: "According to this premise, we are going to investigate IVM.....", which could be changed to: "The effect of IVM on equine and canine immune competent cells such as.... should be investigated". I would also remove the 2 figures, and reserve them for when the results of this study are available. I am are sure to be very interesting, regardless of the direction they finally take (antibacterial action of IVM or not). I would also remove phrases such as the one on line 372 (“The draft of the project and experimental design.....”), 539 (“This branch of our research pipeline will work....”) . I would also change in line 623 "with this research line..." to "these research studies help address..." (or something similar), and the same at the beginning of line 628.
- Ivermectin should be written in lower case except at the beginning of a sentence.
- The bibliographical references should be separated in the text from the word immediately before.
- The scientific names of the microorganisms in the References section should be in italics.
Author Response
Reviewer 3
The word biocide should not be used, as it may be misleading. In the European Union, a clear distinction is made in the legislation between substances that are considered veterinary medicinal products, biocides or feed additives.
Response
Many thanks to the referee for this comment. Now amended as requested.
The symbol µg has not been corrected throughout the manuscript and still appears in several places (e.g. lines 88, 170). And numbers should be separated from units (line 200, 2 mg/kg).
Response
Many thanks to the referee for this comment. Now amended.
I have reviewed those papers published in the journal Veterinary Sciences under the category Perspective, and they present results or review the state of the art of a certain topic. Perhaps the main problem is that the manuscript describes the proposal of a future research project to be carried out. In my experience, what I have seen throughout different journals is that authors submit a research project to certain journals for validation, and then they publish the results in that journal, but they do not publish the research project itself. I think, however, that the review is exhaustive and interesting, and opens a window for research on the role played by substances such as ivermectin in antimicrobial resistance. For this reason, they should approach the paper to be written in a more impersonal way. For example, instead of saying in line 366 "From our perspective, we will pursue the study of....", one could say "From our perspective, the study of the possible induction of macrolides antibiotics resistance in various bacterial species should be studied/assessed". And the same in line 543: "According to this premise, we are going to investigate IVM.....", which could be changed to: "The effect of IVM on equine and canine immune competent cells such as.... should be investigated". I would also remove the 2 figures, and reserve them for when the results of this study are available. I am are sure to be very interesting, regardless of the direction they finally take (antibacterial action of IVM or not). I would also remove phrases such as the one on line 372 (“The draft of the project and experimental design.....”), 539 (“This branch of our research pipeline will work....”) . I would also change in line 623 "with this research line..." to "these research studies help address..." (or something similar), and the same at the beginning of line 628.
Ivermectin should be written in lower case except at the beginning of a sentence.
The bibliographical references should be separated in the text from the word immediately before.
The scientific names of the microorganisms in the References section should be in italics.
Response
All the comments have been fulfilled, hopefully, well and clearly enough. Reading again throughout the manuscript, we realized that it is much better to use the impersonal form and we changed it in every section. We would like to keep the figures in the manuscript, if the referee agrees with that, to provide a graphic idea of the possible experimental design.
I am personally very happy with this referral and comments that provided anytime a correct and objective point of view through all the discussion and that helped to reshape the manuscript in a much better way.